# Principal Component Analysis from Mass Spectrometry Data Combined to a Sensory Evaluation as a Suitable Method for Assessing Bitterness of Enzymatic Hydrolysates Produced from Micellar Casein Proteins

**DOI:** 10.3390/foods9101354

**Published:** 2020-09-24

**Authors:** Dahlia Daher, Barbara Deracinois, Alain Baniel, Elodie Wattez, Justine Dantin, Renato Froidevaux, Sylvie Chollet, Christophe Flahaut

**Affiliations:** 1UMR Transfrontalière BioEcoAgro N° 1158, Univ. Lille, INRAE, Univ. Liège, UPJV, YNCREA, Univ. Artois, Univ. Littoral Côte d’Opale, ICV—Institut Charles Viollette, F-59000 Lille, France; d.daher@ingredia.com (D.D.); barbara.deracinois@univ-lille.fr (B.D.); renato.froidevaux@univ-lille.fr (R.F.); sylvie.chollet@yncrea.fr (S.C.); 2Ingredia S.A. 51 Av. Lobbedez-CS 60946, 62033 Arras Cedex, France; a.baniel@ingredia.com (A.B.); e.wattez@ingredia.com (E.W.); j.dantin@ingredia.com (J.D.)

**Keywords:** biocatalysis, bitterness, micellar casein hydrolysates, off-flavors, peptidomics, sensory analysis

## Abstract

Enzymatic hydrolysis of food proteins generally changes the techno-functional, nutritional, and organoleptic properties of hydrolyzed proteins. As a result, protein hydrolysates have an important interest in the food industries. However, they tend to be characterized by a bitter taste and some off-flavors, which limit their use in the food industry. These tastes and aromas come from peptides, amino acids, and volatile compounds generated during hydrolysis. In this article, sixteen more or less bitter enzymatic hydrolysates produced from a milk protein liquid fraction enriched in micellar caseins using commercially available, food-grade proteases were subjected to a sensory analysis using a trained and validated sensory panel combined to a peptidomics approach based on the peptide characterization by reverse-phase high-performance liquid chromatography, high-resolution mass spectrometry, and bioinformatics software. The comparison between the sensory characteristics and the principal components of the principal component analysis (PCA) of mass spectrometry data reveals that peptidomics constitutes a convenient, valuable, fast, and economic intermediate method to evaluating the bitterness of enzymatic hydrolysates, as a trained sensory panel can do it.

## 1. Introduction

Advances in nutrition have brought to the market high quality industrial products composed of semi-elemental nutrients characterized by a remarkable digestive tolerance, a decrease in allergenicity, a reinforcement of bioavailability and bioactivity, and a good nutritional value. Some of those products are based on protein hydrolysates. Towards the end of the 1970s, the idea that milk proteins could have physiological roles emerged. Indeed, in addition to encrypted bioactive peptides in their sequence [1], hydrolysates have also been found to have a nutritional function: They are more and more used in the nutrition markets such as in infant nutrition, clinical nutrition, and sport nutrition. For example, caseins are frequently used in food formulations because of their high nutritional quality. Following their enzymatic hydrolysis, caseins are used as dietary supplements to promote their assimilation, as well as in specialized diets for adults with digestive tract disorders [2]. In addition, casein hydrolysates are used since 1942 in milks for children suffering with bovine milk allergy [3]. Indeed, the hydrolysates often reduce the protein allergenicity.

The development and marketing of products based on casein enzymatic hydrolysates that have the capacity to provide the desired organoleptic, functional, physiological, and nutritional qualities are a major objective of the dairy industries. Moreover, enzymatic proteolysis is economical, not dangerous, non-polluting, does not cause degradation or racemization of amino acids [4], minimizes the production of undesirable by-products, and allows the preservation of nutritional and functional properties of peptides [5]. However, a bitter taste may appear [6] due to the hydrolysis of dairy proteins and lead to the appearance of off-flavors. For example, cheese or rancid tastes could limit the use of protein hydrolysates because they could cause a taste defect in the products in which they are used. The structural changes associated with the hydrolysis of a protein are: (i) the decrease in the molecular weight of peptide chains, which increases digestibility and helps to reduce their allergenic potential [7], and (ii) the exposure of hydrophobic sites normally localized at the core of the protein structure, which can in turn modify the solubility of peptides.

Tastes of ingested compounds are perceived thanks to four types (types I to IV) of taste bud cells [8], mainly located on the tongue, and the type II cells express, among others, the bitter taste receptors. These latter constitute a family of heptahelical G protein-coupled receptors, called taste 2 receptors (T2Rs). The number of T2R genes differs greatly across mammalian species, and each individual does not have the same sensitivity toward bitterness due to single nuclear polymorphisms [9]. Humans have a least 25 bitterness receptors, among them five concern the bitterness receptors of peptides [10,11]. The number of bitter compounds is estimated to be several hundred for natural compounds alone [12]. Many of them are found in food and beverages, for example naringin in grapefruit, neohesperidin in citrus fruits, caffeine and quinine in some plants, and some amino acids. Several studies have shown that native animal or vegetable proteins are very rarely bitter, whereas their constitutive amino acids and peptides obtained by hydrolysis are very often bitter [13,14]. Indeed, as early as 1909, Ikeda demonstrated that the taste of sake, misa, and soy sauce was presumed to come from the amino acids released from the proteins during fermentation [15]. Subsequently, the taste of cheese was attributed to the amino acids released during cheese ripening [16]. As a result, in 1967, Langler et al. were able to reconstitute a Swiss cheese flavor by combining certain amino acids (L-Pro, Gly, Ser, Thr, Asp, Glu, Cys, Trp, His, Lys) with fatty acids and volatile compounds [17]. In addition to their nutritional role, amino acids therefore play a role in the taste of various foods.

In the 1970s, a lot of research was conducted to define the physico-chemical characteristics of peptides present in hydrolysates associated with bitterness. For example, Matoba et al. showed that there are no bitter peptides for lengths greater than 25 residues [18]. Indeed, in a large peptide, the tertiary structure of the peptide is such that hydrophobic groups are buried inside the molecule to limit contact with aqueous phase and are therefore inaccessible to the bitterness receptors. In a small peptide, however, the hydrophobic side chains are unmasked and can come in contact with the taste receptors [19]. In general, it has been claimed that peptides with a molecular weight less than 6 kDa are responsible for bitterness [20]. It has also been claimed that bitter peptides possess two determinant sites, the binding unit (BU) and the stimulating unit (SU). The BU is a bulky hydrophobic group and the SU is a bulky basic group, including an α- amino group. A hydrophobic group also serves as the SU site for bitter taste. The optimal distance between them is 4.1 Å [19,21].

The bitterness of a peptide can also be related to the hydrophobic nature of its amino acids [22], such as aromatic residues (tryptophan, tyrosine, and phenylalanine) or having a hydrocarbon side chain (leucine, isoleucine, and valine) and their position in the peptide sequence [23]. For example, the Ishibashi works showed that C-terminal leucine, phenylalanine and tyrosine, and/or a proline residue at the geometric center of the peptide intensified its bitterness [24,25]. Siemion et al. also demonstrated that the intensity of bitterness depends on the molecular chirality: peptides with L-configuration amino acids have a higher bitterness than peptides containing D-configuration amino acids [26].

These peptide characteristics are now accepted worldwide and are therefore established according to four major criteria: (i) the size of the peptides (between 100 and 6000 Da), (ii) the nature of their amino acids (aromatic, hydrophobic) [27], (iii) their hydrophobicity expressed as Ney’s coefficient (*Q*-value > to 1400) even if the bitterness of a peptide cannot be predicted simply on the basis of its *Q*-value, and (iv) their structure (diastereoisomer of the L series; proline at the geometric center; proline close to a basic amino acid; hydrophobic amino acids at N- and C-terminal; Tyr, Leu, Phe at the C-terminal end) [1].

Sensomics, a member of a wider family of ’omics´ technologies applied to food, aims to describe sensory properties of foodstuffs at a molecular level [28]. It is a promising instrumental characterization of food, which links analytical data with sensory analysis. The sensomics approach was used to highlight the peptides responsible for the bitterness in gouda cheese [29], Ragusano cheese [30], and whey protein hydrolysates [31]. However, there have never been reported studies using the sensomics approach applied to protein hydrolysates obtained by controlled enzymatic hydrolysis of dairy proteins. Such a sensomics approach represents a very powerful tool to better understand and control the enzymatic processes of protein hydrolysis and to produce peptide hydrolysates while limiting their degree of bitterness.

In the present study, the three aspects of the chosen sensomics methodology are summarized in Figure 1. First, enzymatic hydrolyses were performed from micellar casein proteins using commercially available and food-grade proteases to produce sixteen hydrolysates of varying bitterness. Indeed, several factors, such as the type of enzyme used to hydrolyze the protein and the hydrolysis processing conditions (e.g., pH, temperature, degree of hydrolysis, etc.), influence the taste of the hydrolysates [32]. Second, the organoleptic features of hydrolysates and more particularly bitterness and off-flavors were studied using a trained sensory panel to judge and characterize the produced hydrolysates. Third, hydrolysis generated peptides were characterized by a peptidomics approach combining the peptide chromatographic separation by reverse-phase high-pressure liquid chromatography (RP-HPLC), the detection and fragmentation of peptides by tandem mass spectrometry (MS/MS), and the mass data management using Progenesis QI for proteomics (Waters, Manchester, UK), Peaks studio (Bioinformatics Solution Inc., Waterloo, ON, Canada), and the UniProt-extracted bovine protein database. Those tools lead to characterize the amino acid sequences of identified peptides, to measure their intensity but also to carry out comparative statistical analysis to highlight the qualitative and quantitative relevant differences. Finally, the peptides identified from hydrolysates by peptidomics approach were compared with the sensory characteristics of hydrolysates using appropriate statistical tools, such as the RV coefficient, in order to evaluate the overall similarity of the two matrices and then a correlation test (Pearson) to identify the correlation between the sensory and the analytical data.

## 2. Materials and Methods

### 2.1. Raw Material

Milk protein liquid fraction enriched in micellar caseins (ratio micellar caseins/whey proteins (92:8)) was prepared by the Ingredia S.A. manufacture (St-Pol-Sur-Ternoise, France) using industrial processes. This latter was hydrolyzed with different enzymes alone or in combination. The food grade enzymes Flavourzyme, Protamex, Alcalase, Chymotrypsine were obtained from Novozyme (Bagsvaerd, Denmark), Maxipro PSP and Maxipro FCP were from DSM Food Specialities (Delft, The Netherlands), Prolyve1000, ProlyveBS, LypaÏne were from LYVEN (Collombelles, France), Promod 523MDP, Flavorpro 937MDP were from BIOCATALYSTS (Wales, UK). 

### 2.2. Production of Hydrolysates and Sampling

For the study, sixteen enzymatic hydrolysates were generated using different confidential recipes where the changes concern the nature of enzymes used alone or combined, the enzyme/substrate ratio, and the temperature and pH of hydrolysis. Overall, the protein solution enriched in micellar casein (92%) was diluted with distilled water to a concentration of 10% (expressed in g per 100 g of milk product) of total nitrogenous matter. The latter represents the amount of total nitrogen in the mixture. It corresponds to the proportion of the non-protein nitrogen (urea in majority, amino acid, etc.) and the protein nitrogen. The mixture is brought to the desired pH by adding NaOH (4N) or HCl (4N). The desired enzyme quantity was then added directly if it is in liquid form or solubilized in distilled water if it is in powder form. The hydrolysis monitoring was carried out by collecting data from pH, temperature, osmometry, and the degree of hydrolysis using a method based on the reaction of primary amino groups with o-phthaldialdehyde (OPA). A calibration curve was established at 340 nm with a known concentration of leucine. The degree of hydrolysis is determined by the relationship:
DH = (h/h total) × dilution × 100 (in %)(1)
with h = number of hydrolyzed peptide bonds; h total = total number of mmol of NH_2_ released after total acid chemical hydrolysis measured per 100 mg of micellar casein (h total = 0.6885); dilution of the hydrolysate sample = 100.8
h = ((AH − ANH)/εl)(2)
with AH = absorbance of the sample at hydrolysis time « t »
ANH = absorbance of the sample at t0 before hydrolysis(3)
εl = right slope(4)

After seven to nine hours of hydrolysis, enzymes were inactivated by heating at 98 °C for 3 min and an aliquot of each hydrolysis was frozen at −20 °C before further analytical characterizations. The remaining amount of hydrolysates was dried by atomization using the Mini Spray Dryer B-290 from BUCHI (Rungis, France). The drying process was performed at room temperature (23–25 °C). The dispersions were fed by a peristaltic pump rate between 8 and 13 mL·min^−1^ to maintain the outlet temperature, and the solutions were atomized using 536 L·h^−1^ compressed air. Inlet and outlet air temperatures were 185 °C and 90 °C, respectively. The powder was collected at the bottom of the dryer’s cyclone and stored at room temperature, in a dry place and in the absence of light before to be used for the sensory tests. This dry form allows a better storage and could be a potential product sold to future customers that plan to integrate protein hydrolysates into their various formulations.

Moreover, in order to check the reliability of our method, a commercially available, unknown milk casein hydrolysate (CAMCH) from Arla Foods was purchased online. CAMCH was diluted to a concentration of 10% dry matter in tempered water for 15 min of agitation.

### 2.3. Sensory Analysis

#### 2.3.1. Panel Composition and Training

One group of 19 healthy adults (12 females and 7 males), ranging in age from 45 to 65 years, participated in this study. This group has been recruited based on the NF ISO 22,935-1 standard (July 2009) and trained for 9 months, one hour per week. Assessors were enrolled in a training program designed to form an expert panel in bitterness and off-flavors identification: assessors were trained in order to quantify five odors (milk, fermented milk, rancid, soy milk, smelly), eleven flavors (bitter, sour, milk, sweet, mild, cheese, vanilla, salty, rancid, barn, whey), and five persistence flavors (bitter, sour, milk, sweet, cheese) in the hydrolysates, with a range scale between 1 to 7. Before tasting hydrolysates, for each of the twenty-one descriptors, the performance of the assessors was also checked in terms of discrimination, repeatability, and agreement (Appendix A).

#### 2.3.2. Tasting Conditions and Presentation of Samples

The assessors evaluated the seventeen (16 hydrolysates + CAMCH) samples in a duplicate manner during five sessions under standard sensory conditions (ISO 13,299, 2003). Samples were presented in a sequential monadic way, and their presentation order was different for each assessor and based on a Williams’ Latin-square arrangement. All hydrolysate samples were dissolved in mineral water at a concentration of 10% of dry matter, were presented in white plastic tumblers, and 20 mL was served at room temperature.

#### 2.3.3. Data Analyses

Data were first analyzed using a principal component analysis (PCA) where the variables are the sensory descriptors and the observations are the sixteen hydrolysates. The data were also assessed by a two-way ANOVA considering the products and the consumers as factors and the different descriptor scores as the dependent variable. When a significant effect of product was found, a Duncan’s multiple range test (*p* < 0.05) was performed to compare the products two by two. These statistical analyses were computed with XLStat (XLStat 2020 1.1, Paris, France).

### 2.4. Peptidomics Analysis

#### 2.4.1. Sample Preparation, Peptide Separation, and Peptide Characterization Using HPLC-ESI-qTOF-MS/MS

Aliquot of hydrolysates were thawed overnight at 4 °C, and peptides were purified and concentrated (×25) by C18 solid phase extraction (SPE) using Bond Elut C18 1000 mg mini-columns (Agilent Technologies, Les Ulis, France). Elution of C18-retained peptides was performed with 3 mL of 80% acetonitrile (ACN)/20% water (*v*/*v*), containing 0.1% trifluoroacetic acid (TFA). The peptides purified on Bond Elut C18 mini-columns were dried by centrifugal evaporation (miVac centrifugal vacuum concentrators, Gene Vac, Ipswich, UK) for 3 h at 40 °C. Dried peptides were then dissolved in 200 µL of ultra-pure H_2_O containing 0.1% trifluoroacetic acid (TFA) (*v*/*v*), subjected to 3 vortex/sonication cycles (20 sec/20 sec), centrifuged for 10 min at 9300 g, and the supernatant was collected for separation and identification of peptides.

Ten µL of collected supernatants were chromatographed at 30 °C in reverse-phase ultrahigh-performance liquid chromatography (RP-HPLC) using a C18-AQ column (250 × 3 mm, 3 µm particles, Interchim, Montluçon, France) on a biocompatible ACQUITY UPLC system (Waters, Manchester, UK). Peptide elution was performed, at a rate of 500 µL·min^−1^, according to the following apolar gradient: 1% ACN/0.1% formic acid (FA) (*v*/*v*) for 5 min, then 1 to 10% ACN/0.1% FA (*v*/*v*) for 25 min, then 10 to 30% ACN/0.1% FA (*v*/*v*) for 30 min, and finally 30 to 95% ACN/0.1% FA (*v*/*v*) for 20 min. Collected supernatants of the 17 hydrolysates were analyzed in two series (H1 to H14 + CAMCH and H29 to H36). For each HPLC-MS/MS run session, quality control (QC) samples corresponding to the equivolume mixture of all collected supernatants were injected in three times (at the beginning, in the middle, and at the end of the HPLC-MS/MS analysis session).

Mass spectrometry measurements were made in sensitivity and positive mode using a Synapt-G2-Si-IMS (Waters, Manchester, UK) mass spectrometer running with the proprietary MassLynx software (version 4.1, Waters, Manchester, UK) and previously calibrated using a sodium formate solution. The peptides separated by chromatography were ionized by electrospray using a source voltage of 3 kV and a source desolvation temperature of 300 °C. MS and MS/MS measurements were performed in a data-dependent analysis (DDA) in the measurement range of 50 to 1700 mass-to-charge ratio (*m*/*z*). A maximum of 10 precursor ions with an intensity threshold of 10,000 are selected to be fragmented by collision induced dissociation (CID) at a voltage of 8 to 9 V for low molecular weight ions and 40 to 90 V for high molecular weight ions. Leucine enkephalin ([M+H]^+^ = 556.632 Da) was injected into the system every 2.5 min for 0.5 sec to follow and correct mass spectrometer measurement errors throughout the analysis period.

#### 2.4.2. Bioinformatics for the Treatment of MS Data

The reprocessing of mass spectrometry data and database searches were performed via Peaks Studio version 8.5 (Bioinformatics Solutions Inc., Waterloo, ON, Canada) using the UniProt database (10 September 2018) restricted to the complete proteome of Bos taurus organism. The mass tolerance thresholds for precursor ions and fragments were defined at 35 ppm and 0.2 Da, respectively, and the data searches were performed notifying three missing cleavage sites and without notifying the choice of an enzyme. Variable methionine oxidation was also considered. The relevance of peptide identities was judged according to their identification generated by PEAKS Studio 8.5 (*p* < 0.05) and a false discovery rate (FDR) of strictly less than 1%. 

Raw data from all HPLC-MS/MS runs were also imported in Progenesis QI for proteomics software (Version 4.1, Nonlinear Dynamics, Newcastle upon Tyne, UK) where: (i) data alignment was automatically managed by Progenesis software using one of all QC runs as reference, (ii) a slight manual alignment was performed to optimizing the HPLC-MS/MS run alignment, (iii) all runs were used for peak picking with a peak picking low limit of 5000, a maximum charge of +4, and retention times defined between 5 to 50 min when (iv) chromatographic peak width were not limited, and (v) data normalization was automatically performed for the statistical analysis in main components (PCA). The filtering criteria used for the statistical comparison of mass signals of HPLC-MS/MS runs were set as follow: (i) a minimum intensity threshold of 1e4 and (ii) only those corresponding to peptides identified. The variables used are derived from the comparison of peptide maps, i.e., the position of the isotopic massifs and their intensity. To allow the comparison of the MS data and MS/MS data, a one-way ANOVA considering the MS and MS/MS as factors and the different numbers of MS and MS/MS scan as the dependent variable was done.

## 3. Results

Sixteen enzymatic hydrolysis batches (named H1, H2, H3, H4, H5, H6, H7, H8, H9, H10, H13, H14, H29, H32, H35, and H36) were designed and carried out accordingly to already reported bibliographic literature related milk protein hydrolysis [6,25] using commercially available proteases developed and produced for food industries. In the following part, sensory and mass spectrometry characterization of these 16 hydrolysates will be presented, as well as their comparison. Moreover, an unknown hydrolysate (CAMCH) of milk caseins was analyzed according to the same protocols and in the same conditions, except that the peptide identification by HPLC-MS/MS and bioinformatics was performed first to evaluate in part the reliability of our approach. 

### 3.1. Sensory Results

#### 3.1.1. Principal Component Analysis (PCA)

For PCA, we kept the first five principal components explaining 85.95% of the variance. Here, only the first two principal components are represented and illustrate the hydrolysates (Figure 2b) and the descriptors (Figure 2a). The principal component 1 opposes H36 and H29, which can be defined as mild with a milk taste to H6, H9, and H13, which are, respectively, characterized by a rancid, cheese, and barn flavor, a rancid odor, and a persistence of a cheese taste. The principal component 2 opposes H3, which has a milk/fermented milk odor, to H1, which is bitter with a persistence of the bitterness. H2 and H5 have a whey and sweet taste contrary to H35 and H32. H10, which is opposite to H14 and H8, is described as sour with a soy milk odor. H4 has a whey taste and H7 can be defined as mild. 

#### 3.1.2. ANOVA Analyses

A finer analysis was made using ANOVA and Duncan’s multiple range tests. This descriptive sensory evaluation provides us a detailed profile of the different product’s sensory attributes as well as a quantitative measurement of the attribute’s intensity (Table 1). Results show that hydrolysates are discriminated for all descriptors (*p* < 0.05). Among the 21 descriptors, a focus was made on the descriptors bitter, bitter persistence, and some defects such as cheese, rancid, and barn. Concerning the descriptor bitter, four hydrolysates (H3, H14, H29, and H36) stand out from the batch being significantly less bitter (*p* < 0.0001) than the others. Concerning the persistence of bitterness in the mouth, three hydrolysates (H3, H36, and H29) tend to be less persistent. Particular attention is paid to hydrolysate H1, the bitterest hydrolysate, which is the bitterness reference in the company. Regarding the previously mentioned taste defects, they also appear to be an obstacle to the use of hydrolysates in the food industry. Among the four less bitter hydrolysates, H29 and H36 are considered as the ones that have the less strong off-flavors (H29: rancid: 1.875; barn: 1.542; cheese: 1.542/H36: rancid: 1.458; barn: 1.333; cheese: 1.542). The new recipes applied on the micellar caseins have therefore resulted in hydrolysates that are less bitter than the reference used in the company with two of them having very low off-flavor intensities (H36 and H29).

### 3.2. Peptide Characterization by Peptidomics Approach and Bioinformatics Treatment of Raw Data

Concentrated and desalted peptides of each hydrolysate were re-solubilized and subjected to an HPLC-MS/MS analysis on a C18-AQ chromatographic column, concomitantly to quality control. MS and MS/MS data were acquired using a high-resolution mass spectrometer. The number of mass spectra scans (MS) ranged from 2949 for hydrolysate H1 to 5118 for hydrolysate H36 with a mean of 4730 MS scans and a standard deviation (SD) of 638. The MS/MS scans ranged from 8721 for hydrolysate H13 to 16,379 for hydrolysate H1 for a mean ± SD of 13,131 ± 1241, respectively. The ANOVA test performed with these values demonstrates that there is no statistically significant differences (*p* = 0.62) between the mass spectrometry data collected from each HPLC-MS/MS run. 

The HPLC-MS/MS raw data obtained for the 17 hydrolysates and QCs were imported in Progenesis QI for proteomics software (version 4.1, Waters Corporation, Milford, MA, USA), as described in the Materials and Methods Section. Among the 2929 peak picked mass signals, 2024 mass signals have an intensity threshold > to 1e4 and among them, 208 mass signals were identified as peptides (123 from beta-casein, 35 from alpha-S1 casein, 22 from alpha-S2 casein, 19 from kappa-casein, and nine from beta-lactoglobulin) with certainty. No peptides from alpha-lactalbumin were identified. These 208 peptides represent the global diversity (all hydrolysates combined) of identified peptides where between 169 and 205 peptides were identified per hydrolysate. The peptide length is comprised between five and 30 amino acids with a length mean of 11 amino acids. The mean of the peptide molecular mass is 1176.48 ± 336.76 Da. A limitation to note is that only peptides that were between 500 to 3000 Da were identified in this analytical context. The identified peptides of β-casein corresponded mainly to five protein regions: T56-T70, Q87-V107- A116-E136, S181-E205, and L207-V224. The α-S1 casein- and α-S2 casein-peptides corresponded to 2 (L113-E125 and L184-W214) and 1 (L114-N130) protein regions, respectively. The kappa-casein- and β-lactoglobulin peptides corresponded to 1 (A117-S125) and 2 (K63-L73 and I87-V97) protein regions, respectively. 

The object of differential analysis is to find those peptides that show abundance difference between experiment groups or individuals. Progenesis QI uses ANOVA tests to formalize the abundance difference. The tests return a *p*-value that takes into account the mean difference and the variance and also the sample size. The PCA in Progenesis QI uses the compound abundance levels across runs to determine the principle axes of abundance variation. Thus, the first five principal components of the PCA explaining 72.78% of the variance were retained. The following PCA was done using the 208 identified peptides (Figure 3a). Only the first two principal components are represented and illustrate the correlations between the sixteen hydrolysates (colored spots on the plot). The more distant the groups, the more different in terms of peptide population.

Figure 3a displays the PCA generated using the two first principal components (PC1 = 28.98% and PC2 = 13.61%) of filtered peptides where three clusters appear as different: the more distant the groups are, the more different they are in terms of peptide population. Note that color circles are hand-marked to facilitate understanding and have no statistical significance. The red-circled cluster (on the left) gathers the hydrolysates H1; H2, H4, H5, H6, H7, H8, H9, H10, and H14. The blue central cluster gathers the hydrolysates H13, H32, H35, and H36, while the green-circled cluster (on the left) gathers only hydrolysates H3 and H29. Figure 3b depicts the PCA corresponding to the 32 peptides most abundant in H3 than in other hydrolysates (H13, H2, H14, H5, H1, H32, H4, H29, H36, and CAMCH) and demonstrates that these latter peptides clearly allow us to statistically distinguish the hydrolysate H3 from other hydrolysates.

### 3.3. Relationship between Sensory and Analytical Data

The relationship between the two sets of variables (the first five principal components of the PCA related to mass spectrometry data and the 21 descriptors of sensory analysis) has been evaluated by the RV coefficient calculated to 0.369 (*p*-value = 0.020). This probability lower than 5% indicates that there are more similarities than dissimilarities between mass spectrometry data and sensory data. 

To better highlight the common features between these two sets of data, Pearson’s coefficient was computed between each principal component of mass spectrometry PCA and each of the 21 sensory descriptors. As summarized in Table 2, the first principal component of PCA related to mass spectrometry data is negatively correlated with the descriptors acid, bitter, acid persistence, bitterness persistence, and vanilla and positively correlated with mild, smelly, milk, milk odor, and milk persistence. The third principal component is negatively correlated with the descriptors cheese persistence and salty, and the fifth principal component is positively correlated with the descriptors smelly, fermented milk odor, and soya milk odor. For the second and fourth principal component, no correlation was found.

It is interesting to note that, CAMCH appeared on the PCA graph (Figure 3a) in the group consisting of H1, H2, H4, H7, etc., which is in the negative part of the principal component 1. Therefore, the CAMCH hydrolysate could be qualified as sour, bitter, sour persistence, etc. Otherwise, the position of CAMCH on the PCA graph (Figure 3a) is at the opposite end of the most bitter hydrolysate H1 (Table 1) suggesting that CAMCH is less bitter than the hydrolysates close to the position of H1 on the PCA graph. Concomitantly, CAMCH was characterized by the sensory panel as sour with a persistence of this sourness and a slight milk odor.

## 4. Discussion

The sensory evaluation carried out by the trained sensory panel on the 16 hydrolysates revealed four products (H3, H14, H29, and H36) less bitter than H1, the bitterest reference hydrolysate. Among these, two have very few off-flavors (H29 and H36). Briefly, it could be explained by the effectiveness of the enzymes and/or enzyme cocktails used to reduce the bitterness of the hydrolysates. Indeed, among the latter, some studies have proven their effectiveness in debittering hydrolysates [33,34]. In addition, some commercially available and food-grade enzymes or enzyme cocktails have proven their efficiency through supplier-specific in-house tests. 

Concomitantly, the 16 hydrolysates were analyzed by RP-HPLC-ESI-MS/MS and a PCA had been performed from peptides identified within the hydrolysates. PCA is a multivariate technique that analyzes a data table in which observations are described by several inter-correlated quantitative dependent variables. The PCA goal is (i) to extract the important information from the table, (ii) to represent it as a set of new orthogonal variables called principal components, and (iii) to display the pattern of similarity of the observations and of the variables as points in maps [35]. Note that the RV coefficient (*p* = 0.020) demonstrates similarities between the five principal components of the PCA and the sensory data. Three groups can be identified from the PCA. The group composed of the greatest number of hydrolysates among them H1, the bitterest hydrolysate with a strongest bitterness persistence. A group of the four hydrolysates H13, H32, H35, and H36 were identified, and among them, H36 is considered by the sensory panel as one of the least bitter hydrolysates. The third group is composed of only two hydrolysates H3 and H29, which are perceived as less bitter. Taking into account that H1 is the bitterest hydrolysate and H3/H29 the less bitter hydrolysates, the PCA illustrates that the H3 and H29 hydrolysates are the opposite of the bitterest hydrolysate. These opposite positions on the PC1-versus-PC2 diagram of H1 and H3/H29 correlate with the bitterness degree of hydrolysate reported by the sensory panel. Furthermore, the hydrolysates having a mean value of bitterness between those of H1 and H3 (Table 1) are positioned on the PC1-versus-PC2 diagram between the hydrolysates of H1 group and those of H3 group. However, H14 is categorized as less bitter by the sensory panel, but H14 belongs to the first group. This fact can be explained by its marked vanilla taste. This descriptor is negatively correlated with the first principal component (Table 2). H14 is also characterized by the absence of milk-like smell, and this milk odor descriptor is positively correlated with the PC1.

An evaluation of this approach was done by the blind HPLC-MS/MS and sensorial analyses of an unknown hydrolysate (CAMCH) in order to predict the taste profile of this hydrolysate. The analyses of CAMCH allow us to validate in part the reliability of such an approach. Consequently, this approach will be used to sort the hydrolysates and to select the most relevant hydrolysates according to the industrial goal. As we know, the development of innovative hydrolysates from a protein mixture enriched in milk caseins is a time-consuming process due to various parameters related to the industrial process. Indeed, even if certain factors can have a less impact than other, the large number of factors (pH, temperature, hydrolysis time, degree of hydrolysis, enzyme/substrate ratio, enzyme combination, etc.) involved in the enzymatic hydrolysis of proteins multiplies with the number of hydrolysates that will be generated to find the adequate one for a given industrial applications. In addition, the industrial process used to condition the hydrolysates are additional parameters. Due to the number of steps in the development of an innovative hydrolysate, the systematic use of sensory panel to characterizing the organoleptic features of in laboratory-produced hydrolysates is unconceivable for practical and economic reasons. Indeed, to explain it further, since humans do not always display the same sensitivity towards bitterness and even if the electronic tongues are an aid [23,36], the end-user test of bitterness based on a trained sensory panel are the only valuable tests. However, the sensory evaluations have several drawbacks such as the fact that it allows only the sensory test of a few products per session, the need of the selection, training, validation, and maintenance of sensory panels, which is time-consuming and expensive. Therefore, the setup of an intermediate statistical method able to evaluate the bitterness of protein hydrolysates as a trained sensory panel can do it through a conventional sensory analysis is a valuable aid for the development of new non-bitter hydrolysates. 

This tool, which is still perfectible, will therefore serve to focus attention on the least bitter hydrolysates and allow us to deepen our research on the causes of this bitterness. Indeed, it would be interesting to understand why they are less bitter than the others in terms of peptide structure, peptide length, hydrophobicity score, etc. In fact, the features of the peptide bitterness are becoming more and more defined, but some points still remain controversial. Indeed, the size of bitter peptides is generally comprised between 100 and 6000 Da. The di- and tripeptides are also responsible of peptide bitterness [27,37,38,39]. However, the empirical correlation between the absence or presence of bitterness and the average of hydrophobicity termed the Q-value does not take into account the peptide net charge, and finally, the bitterness of peptides is mainly evaluated according to the hydrophobicity of amino acids that composed the peptide sequence [19,29]. It is now generally accepted that the side-chain hydrophobicity and the number of carbon atoms of the hydrophobic amino acid side chain are correlated to bitterness rather than to overall hydrophobicity [19]. Chirality and location of key amino acids into the amino acid sequence are also crucial parameters [24,25,26] especially because amino acid racemization can occur during industrial processes. However, until now, even with all these molecular signatures, the bitterness score of peptides is not a calculable value. Computer-assisted evaluation of peptide bitterness has started with the publication of Ney and Retzlaff [40]. A computational method recently published to discriminate bitter peptides from non-bitter peptides will perhaps soon demonstrate its efficiency [41].

## 5. Conclusions

The development of protein hydrolysates, especially casein enzymatic hydrolysates, which display desired organoleptic, functional, physiological, and nutritional properties is a permanent challenge for the food industries due to the multiple parameters involved in their design, manufacture, and industrialization. One of limiting factors of casein hydrolysates that hinders their industrial applications is their bitterness. The bitterness is not perceived with the same intensity for all individuals and is a taste often less appreciated than others. Therefore, the organoleptic evaluation of a protein hydrolysate requires the set-up of normed sensory evaluation sessions with a trained sensory panel. However, the drawbacks of these latter hinder, somewhat, the innovation related to protein hydrolysates. Consequently, a reliable, statistically validated, fast, and robust method to evaluate the bitterness of hydrolysates as a sensory panel can do it would be a valuable method to accelerate the protein hydrolysate development. To our knowledge, there have never been reported studies using a sensomics approach applied to protein hydrolysates obtained by controlled enzymatic hydrolysis of dairy proteins.

Here, we demonstrate that peptidomics data obtained by RP-HPLC-MS/MS analysis of peptides and sensory data obtained thanks to a trained sensory panel have more similarities than dissimilarities and can then be confronted with each other. The statistical analyses of sensory data highlighted a group of four hydrolysates significantly less bitter (*p* < 0.05) than the 12 other hydrolysates and with less off-flavors, such as rancid, etc. Further, three principal components (PC1, PC3 and PC5) of PCA generated from peptides identified by mass spectrometry are correlated with some sensory descriptors, such as bitterness, bitterness persistence, or milk persistence. Finally, the PCA of the more relevant mass spectrometry data appears as a convenient, reliable, fast, and economic intermediate method to evaluating the bitterness of enzymatic hydrolysates as a trained sensory panel can do it. This latter point was confirmed by blind HPLC-MS/MS analysis and blind sensory test of commercially available, unknown milk casein hydrolysate.

## Figures and Tables

**Figure 1 foods-09-01354-f001:**
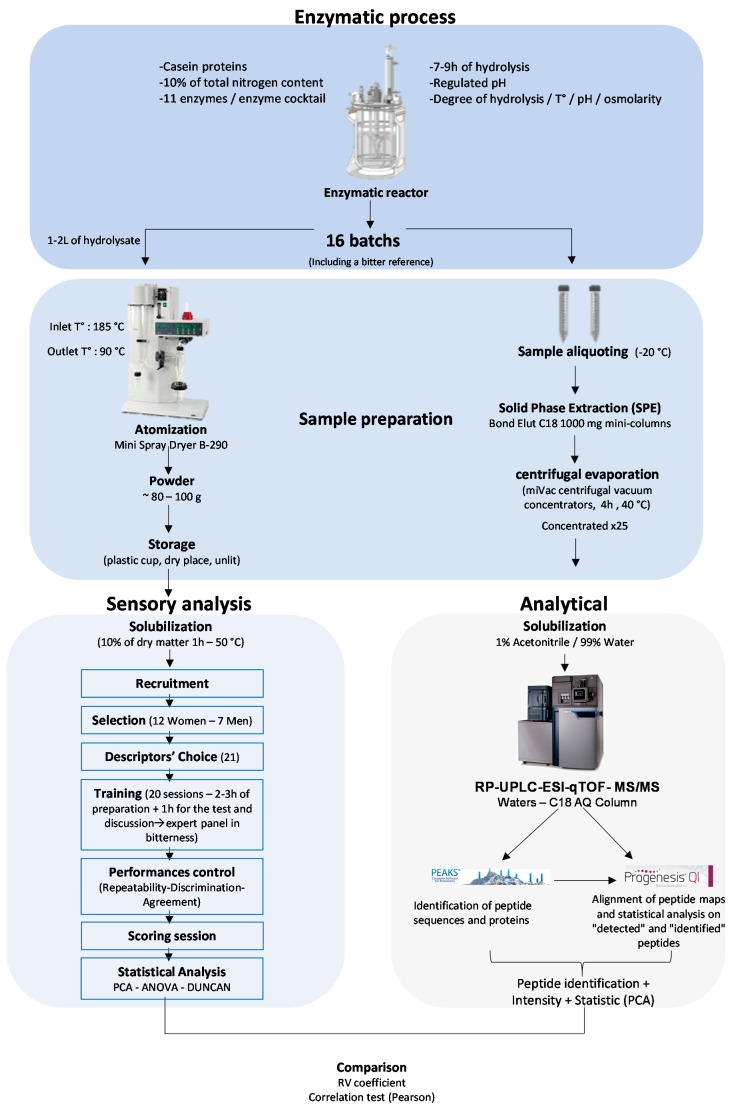
Summary outline of the experimental strategy involving the enzymatic process, sensory evaluation, and mass spectrometry analysis. Principal component analysis (PCA).

**Figure 2 foods-09-01354-f002:**
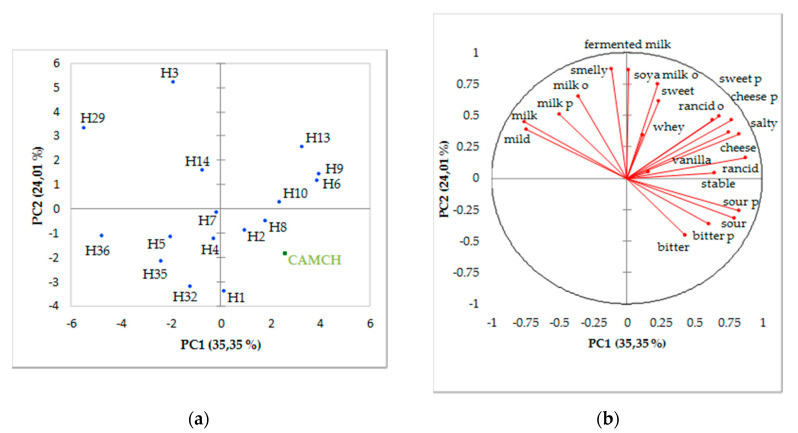
Principal component analysis (PCA) and correlation circle for principal components 1 and 2: (**a**) hydrolysates of milk protein liquid fraction enriched in micellar caseins (ratio micellar caseins/whey proteins (92:8)); (**b**) hydrolysate descriptors. A commercially available milk casein hydrolysate (CAMCH) was projected as supplementary observation.

**Figure 3 foods-09-01354-f003:**
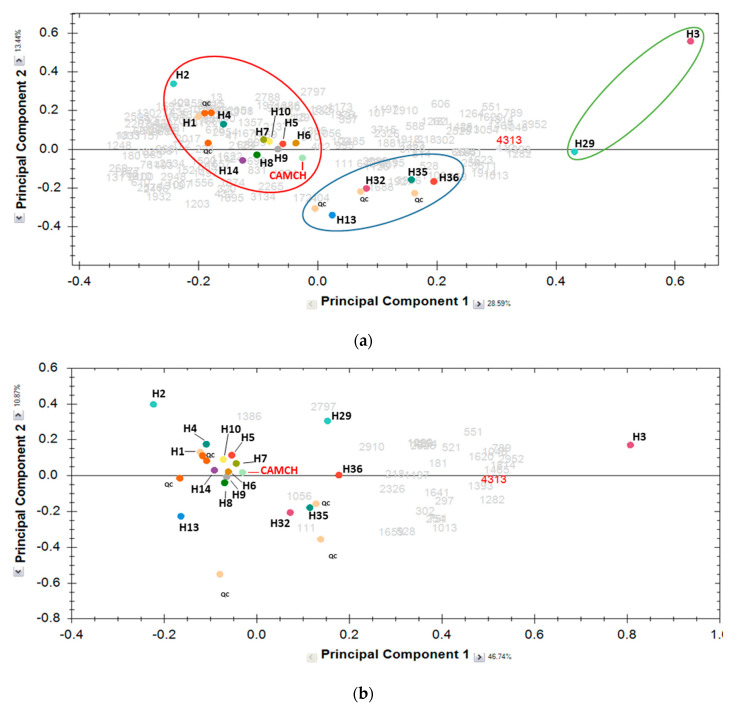
Principal component analysis of the most abundant identified peptides corresponding to the commercially available, unknown milk casein hydrolysate (CAMCH) and the 16 hydrolysates of (**a**) milk protein liquid fraction enriched in micellar caseins (ratio micellar caseins/whey proteins (92:8)) and (**b**) the 32 peptides (depicted in grey numbers) most abundant in H3 than in other hydrolysates (H13, H2, H14, H5, H1, H32, H4, H29, H36, and CAMCH). Principal components #1 and #2 of A explain 28.92% and 13.56% of variance, while principal components #1 and #2 of B explain 47.12% and 10.98% of variance, respectively. Grey numbers correspond to identified peptides displaying a mass signal intensity > to 1e4, while red number corresponds to one peptide of the H3 hydrolysate displaying the greatest fold change (infinity). Color circles are hand-marked to facilitate understanding and have no statistical significance.

**Table 1 foods-09-01354-t001:** Sensory analysis of the seventeen hydrolysates including the Duncan’s multiple range test ^a^.

	Descriptors	H1	H2	H3	H4	H5	H6
ODOR	milk	2.02 ± 1.41 ^cde^	1.99 ± 1.50 ^cde^	2.72 ± 1.60 ^ab^	1.99 ± 1.53 ^cde^	2.09 ± 1.44 ^cde^	1.96 ± 1.19 ^de^
	fermented milk	1.57 ± 0.98 ^bcd^	1.67 ± 1.23 ^bcd^	2.27 ± 1.78 ^a^	1.74 ± 1.56 ^abcd^	1.53 ± 0.71 ^cd^	1.87 ± 1.30 ^abcd^
	rancid	1.49 ± 0.94 ^e^	1.53 ± 1.02 ^de^	2.23 ± 1.75 ^abc^	1.90 ± 1.77 ^bcde^	1.53 ± 1.02 ^de^	2.63 ± 1.95 ^a^
	soya milk	1.34 ± 0.67 ^cd^	1.61 ± 1.41 ^abcd^	1.84 ± 1.53 ^ab^	1.51 ± 1.06 ^abcd^	1.41 ± 0.89 ^abcd^	1.57 ± 0.98 ^abcd^
	smelly	1.92 ± 1.43 ^e^	1.85 ± 1.27 ^e^	3.02 ± 1.47 ^ab^	2.22 ± 1.29 ^e^	1.98 ± 1.49 ^e^	2.88 ± 1.70 ^abc^
FLAVORS	bitter	5.68 ± 1.24 ^a^	5.01 ± 1.58 ^bc^	3.31 ± 1.84 ^e^	4.81 ± 1.58 ^bcd^	4.58 ± 1.74 ^bcd^	4.78 ± 1.71 ^bcd^
	sour	2.67 ± 1.95 ^cdef^	2.93 ± 2.16 ^bcd^	1.97 ± 1.08 ^g^	2.90 ± 1.87 ^bcd^	2.80 ± 1.67 ^bcde^	3.23 ± 1.97 ^bc^
	milk	1.84 ± 1.32 ^bcd^	2.01 ± 1.49 ^bcd^	2.34 ± 1.47 ^ab^	2.01 ± 1.52 ^bcd^	2.34 ± 1.79 ^ab^	1.77 ± 1.04 ^cd^
	mild	1.10 ± 0.19 ^e^	1.30 ± 0.77 ^e^	1.97 ± 1.45 ^ab^	1.20 ± 0.46 ^e^	1.80 ± 1.37 ^abc^	1.13 ± 0.27 ^e^
	sweet	1.39 ± 1.06 ^ab^	1.59 ± 1.35 ^ab^	1.69 ± 1.48 ^a^	1.39 ± 1.23 ^ab^	1.45 ± 1.24 ^ab^	1.55 ± 1.44 ^ab^
	rancid	2.24 ± 1.77 ^de^	2.27 ± 1.73 ^de^	1.91 ± 1.61 ^ef^	2.07 ± 1.48 ^def^	1.54 ± 0.75 ^f^	3.24 ± 2.19 ^a^
	whey	1.68 ± 1.05 ^bc^	2.02 ± 1.50 ^abc^	2.18 ± 1.39 ^a^	2.12 ± 1.61 ^ab^	2.02 ± 1.45 ^abc^	1.92 ± 1.44 ^abc^
	barn	1.80 ± 1.33 ^b^	1.73 ± 1.25 ^b^	1.67 ± 1.29 ^b^	1.63 ± 0.94 ^b^	1.53 ± 0.70 ^b^	2.87 ± 2.03 ^ab^
	cheese	1.80 ± 1.36 ^cd^	1.77 ± 1.25 ^cd^	1.94 ± 1.23 ^cd^	1.97 ± 1.21 ^cd^	1.70 ± 1.23 ^d^	2.54 ± 1.79 ^ab^
	vanilla	1.52 ± 1.20 ^abc^	1.65 ± 1.28 ^ab^	1.42 ± 1.02 ^abc^	1.39 ± 0.89 ^bc^	1.42 ± 0.90 ^abc^	1.32 ± 0.73 ^bc^
	salty	2.37 ± 2.07 ^abc^	2.81 ± 2.03 ^abc^	2.57 ± 1.65 ^abc^	2.24 ± 1.98 ^c^	2.64 ± 2.24 ^abc^	2.61 ± 2.24 ^abc^
PERSISTENCE OF FLAVORS	bitter	4.78 ± 1.78 ^a^	4.65 ± 1.55 ^ab^	2.85 ± 1.51 ^g^	4.42 ± 1.52 ^abc^	4.18 ± 1.70 ^abcd^	4.02 ± 1.45 ^bcd^
	sour	2.65 ± 1.99 ^bcd^	2.65 ± 1.87 ^bcd^	1.95 ± 1.07 ^fg^	2.35 ± 1.54 ^cdef^	2.22 ± 1.47 ^def^	2.72 ± 1.77 ^bcd^
	milk	1.97 ± 1.45 ^abc^	2.07 ± 1.69 ^abc^	2.23 ± 1.61 ^ab^	2.03 ± 1.61 ^abc^	2.17 ± 1.85 ^ab^	1.90 ± 1.33 ^abc^
	sweet	1.35 ± 1.05 ^a^	1.49 ± 1.20 ^a^	1.49 ± 1.02 ^a^	1.39 ± 1.23 ^a^	1.42 ± 0.90 ^a^	1.42 ± 1.23 ^a^
	cheese	1.80 ± 1.55 ^efg^	1.90 ± 1.24 ^defg^	2.10 ± 1.53 ^cdefg^	1.77 ± 1.15 ^fg^	1.67 ± 1.16 ^g^	2.40 ± 1.98 ^abc^
	**Descriptors**	**H7**	**H8**	**H9**	**H10**	**H13**	**H14**
ODOR	milk	2.02 ± 1.38 ^cde^	1.85 ± 1.24 ^e^	2.52 ± 1.50 ^bc^	2.45 ± 1.74 ^bcd^	2.18 ± 1.46 ^cde^	1.98 ± 1.40 ^cde^
	fermented milk	1.91 ± 1.54 ^abcd^	1.61 ± 1.33 ^bcd^	2.11 ± 1.70 ^ab^	1.78 ± 1.42 ^abcd^	1.95 ± 1.22 ^abcd^	1.98 ± 1.64 ^abcd^
	rancid	1.96 ± 1.22 ^bcde^	2.16 ± 1.73 ^abcde^	2.33 ± 1.71 ^ab^	1.96 ± 1.25 ^bcde^	2.63 ± 1.70 ^a^	2.20 ± 1.64 ^abcd^
	soya milk	1.52 ± 1.01 ^abcd^	1.52 ± 1.15 ^abcd^	1.69 ± 1.10 ^abc^	1.86 ± 1.57 ^a^	1.66 ± 1.17 ^abc^	1.49 ± 1.01 ^abcd^
	smelly	2.24 ± 1.21 ^de^	2.14 ± 1.15 ^e^	2.40 ± 1.18 ^cde^	2.34 ± 1.44 ^cde^	2.84 ± 1.42 ^abcd^	2.47 ± 1.52 ^bcde^
FLAVORS	bitter	4.67 ± 1.58 ^bcd^	4.33 ± 1.64 ^cd^	4.50 ± 1.71 ^bcd^	5.10 ± 1.64 ^ab^	4.57 ± 1.59 ^bcd^	3.37 ± 1.43 ^e^
	sour	2.71 ± 1.53 ^bcdef^	2.78 ± 2.17 ^bcde^	3.41 ± 1.94 ^b^	3.34 ± 1.75 ^bc^	2.81 ± 1.46 ^bcde^	2.14 ± 1.14 ^efg^
	milk	2.14 ± 1.51 ^bc^	2.14 ± 1.32 ^bc^	1.91 ± 1.18 ^bcd^	1.97 ± 1.50 ^bcd^	2.01 ± 1.35 ^bcd^	2.04 ± 1.62 ^bc^
	mild	1.42 ± 1.04 ^de^	1.35 ± 0.62 ^de^	1.08 ± 0.19 ^e^	1.32 ± 0.82 ^de^	1.32 ± 0.54 ^de^	2.16 ± 1.90 ^a^
	sweet	1.44 ± 1.05 ^ab^	1.37 ± 0.83 ^ab^	1.37 ± 1.21 ^ab^	1.31 ± 0.77 ^ab^	1.47 ± 1.12 ^ab^	1.41 ± 1.04 ^ab^
	rancid	2.56 ± 1.78 ^bcd^	2.60 ± 1.73 ^bcd^	3.06 ± 2.24 ^ab^	2.60 ± 1.73 ^bcd^	2.90 ± 1.99 ^abc^	2.03 ± 1.27 ^def^
	whey	1.79 ± 1.33 ^abc^	1.86 ± 1.35 ^abc^	1.96 ± 1.25 ^abc^	1.76 ± 1.09 ^abc^	1.89 ± 1.18 ^abc^	1.89 ± 1.42 ^abc^
	barn	1.80 ± 1.42 ^b^	3.63 ± 1.30 ^a^	2.30 ± 1.88 ^b^	2.07 ± 1.70 ^b^	2.37 ± 1.75 ^ab^	1.73 ± 1.13 ^b^
	cheese	1.97 ± 1.72 ^cd^	2.17 ± 1.52 ^bc^	2.67 ± 2.01 ^a^	2.20 ± 1.80 ^bc^	2.50 ± 1.64 ^ab^	2.24 ± 1.43 ^bc^
	vanilla	1.57 ± 1.39 ^abc^	1.41 ± 0.88 ^bc^	1.27 ± 0.81 ^bc^	1.37 ± 0.96 ^bc^	1.37 ± 0.96 ^bc^	1.77 ± 1.35 ^a^
	salty	2.46 ± 2.11 ^abc^	2.59 ± 2.25 ^abc^	2.99 ± 2.26 ^a^	2.82 ± 2.14 ^abc^	2.96 ± 1.83 ^ab^	2.625 ± 1.73 ^abc^
PERSISTENCE OF FLAVORS	bitter	4.19 ± 1.64 ^abcd^	3.82 ± 1.68 ^cde^	4.45 ± 1.57 ^abc^	4.65 ± 1.48 ^ab^	4.39 ± 1.64 ^abc^	3.25 ± 1.49 ^efg^
	sour	2.20 ± 1.61 ^def^	2.54 ± 1.76 ^bcde^	2.80 ± 1.80 ^bc^	2.94 ± 1.61 ^b^	2.54 ± 1.47^bcde^	1.84 ± 1.01 ^fg^
	milk	2.17 ± 1.56 ^ab^	1.97 ± 1.25 ^abc^	1.80 ± 1.18 ^bc^	2.00 ± 1.46 ^abc^	2.13 ± 1.48 ^abc^	1.97 ± 1.58 ^abc^
	sweet	1.41 ± 1.11 ^a^	1.41 ± 1.00 ^a^	1.44 ± 1.21 ^a^	1.54 ± 1.33 ^a^	1.54 ± 1.39 ^a^	1.44 ± 1.05 ^a^
	cheese	1.84 ± 1.54 ^efg^	2.34 ± 1.75 ^abcd^	2.60 ± 1.96 ^ab^	2.20 ± 1.84 ^bcdef^	2.70 ± 1.8 2 ^a^	2.27 ± 1.58 ^abcde^
	**Descriptors**	**H29**	**H32**	**H35**	**H36**	**CAMCH**	
ODOR	milk	3.17 ± 1.79 ^a^	2.17 ± 1.68 ^cde^	2.07 ± 1.30 ^cde^	1.97 ± 0.97 ^de^	1.78 ± 1.43 ^e^	
	fermented milk	2.02 ± 1.55 ^abc^	1.62 ± 1.38 ^bcd^	1.79 ± 1.28 ^abcd^	1.56 ± 0.88 ^bcd^	1.41 ± 0.88 ^d^	
	rancid	1.63 ± 1.25 ^cde^	1.96 ± 1.51 ^bcde^	1.63 ± 1.06 ^cde^	1.63 ± 0.97 ^cde^	2.20 ± 1.73 ^abcd^	
	soya milk	1.78 ± 1.40 ^abc^	1.14 ± 0.20 ^d^	1.51 ± 1.06 ^abcd^	1.34 ± 0.69 ^bcd^	1.62 ± 1.21 ^abcd^	
	smelly	3.11 ± 1.36 ^a^	2.11 ± 1.26 ^e^	2.21 ± 0.91 ^e^	2.34 ± 1.12 ^cde^	2.14 ± 1.50 ^e^	
FLAVORS	bitter	3.47 ± 1.39 ^e^	4.77 ± 1.56 ^bcd^	4.30 ± 1.75 ^d^	3.33 ± 1.41 ^e^	3.17 ± 1.74 ^e^	
	sour	1.82 ± 1.01 ^g^	2.69 ± 1.65 ^cdef^	2.25 ± 1.39 ^defg^	2.05 ± 1.26 ^fg^	4.41 ± 1.96 ^a^	
	milk	2.71 ± 1.48 ^a^	2.05 ± 1.42 ^bc^	1.91 ± 1.11 ^bcd^	2.35 ± 1.74 ^ab^	1.51 ± 0.89 ^d^	
	mild	1.93 ± 1.36 ^ab^	1.27 ± 0.41 ^e^	1.70 ± 1.18 ^bcd^	1.93 ± 1.60 ^ab^	1.45 ± 1.25 ^cde^	
	sweet	1.38 ± 0.64 ^ab^	1.18 ± 0.44 ^b^	1.28 ± 0.50 ^b^	1.28 ± 0.53 ^ab^	1.37 ± 0.74 ^ab^	
	rancid	1.96 ± 1.51 ^ef^	2.23 ± 1.64 ^de^	1.89 ± 1.31 ^ef^	1.63 ± 0.93 ^f^	2.46 ± 1.82 ^cde^	
	whey	1.68 ± 0.82 ^bc^	1.64 ± 1.24^c^	1.81 ± 1.17 ^abc^	1.94 ± 1.16 ^abc^	1.92 ± 1.14 ^abc^	
	barn	1.63 ± 0.83 ^b^	2.10 ± 1.62 ^b^	1.77 ± 1.08 ^b^	1.47 ± 0.76 ^b^	2.00 ± 1.66 ^b^	
	cheese	1.64 ± 0.72 ^d^	1.90 ± 1.51 ^cd^	1.80 ± 1.42 ^cd^	1.64 ± 0.93 ^d^	1.94 ± 1.49 ^cd^	
	vanilla	1.28 ± 0.60 ^bc^	1.25 ± 0.50 ^c^	1.22 ± 0.48 ^c^	1.28 ± 0.53 ^bc^	1.44 ± 0.84 ^abc^	
	salty	2.31± 1.87 ^c^	2.35 ± 1.85 ^bc^	2.31 ± 2.00 ^c^	2.21 ± 1.3 ^c^	2.66 ± 1.94 ^abc^	
PERSISTENCE OF FLAVORS	bitter	2.81 ± 1.53 ^g^	3.98 ± 1.44 ^bcd^	3.54 ± 1.71 ^def^	2.84 ± 1.31 ^g^	3.09 ± 1.73 ^fg^	
	sour	1.54 ± 0.56 ^g^	1.64 ± 1.02 ^g^	2.04 ± 1.12 ^efg^	1.64 ± 0.83 ^g^	3.80 ± 1.84 ^a^	
	milk	2.34 ± 1.76 ^a^	1.97 ± 1.54 ^abc^	1.74 ± 1.09 ^bc^	2.07 ± 1.55 ^abc^	1.63 ± 0.97 ^c^	
	sweet	1.31 ± 0.69 ^a^	1.25 ± 0.50 ^a^	1.18 ± 0.44 ^a^	1.28 ± 0.53 ^a^	1.47 ± 1.31 ^a^	
	cheese	1.64 ± 1.02 ^g^	1.81 ± 1.56 ^efg^	1.81 ± 1.56 ^efg^	1.71 ± 1.34 ^g^	1.90 ± 1.52 ^defg^	

^a^ Values from 1 to 7 are means +/− standard deviation (*n* = 2): different superscript letters (a, b, c, etc.) indicate means that significantly differ among hydrolysates for each descriptor at *p* < 0.05 according to the Duncan’s multiple range test. Hydrolysate (H) and commercially available milk casein hydrolysate (CAMCH).

**Table 2 foods-09-01354-t002:** Correlation matrix (Pearson’s test).

Principal Components	PC1	PC2	PC3	PC4	PC5
+	milk (0.80) mild (0.63) milk persistence (0.60) milk odor (0.71) smelly (0.53)	/	/	/	smelly fermented milk odor (0.53) soya milk odor (0.57)
−	bitter (−0.73) sour (−0.65) bitter persistence (−0.80) sour persistence (−0.63) vanilla (−0.59)	/	cheese persistence (−0.58) salty (−0.55)	/	/

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
