# Peer review of "Principal Component Analysis from Mass Spectrometry Data Combined to a Sensory Evaluation as a Suitable Method for Assessing Bitterness of Enzymatic Hydrolysates Produced from Micellar Casein Proteins"

_foods, 2020, doi:10.3390/foods9101354_

Round 1

Reviewer 1 Report

Thanks to the authors, the manuscript reads much better now and all points raised by the reviewers have been in my view well addressed. I would still recommend to please include the more recent work of O'Sullivan group in UCD Dublin, Ireland on the subject of peptide bitterness versus chromatographic behaviour as most if not all these references are old. In particular, please review the similar approaches to the present study taken in Talanta 126, 46-53 (2014) and in Journal of Dairy Science 101(4), 2826-2837 (2018)

Some few remaining English language / typos could be looked at before final publication such as: 

Line 51: "cheese or rancid tastes could"

Line 111: "to producing sixteen hydrolysates of varying bitterness"

Line 140: "In Overall, ..."

Line 155: After seven hours of hydrolysis... slight contradiction with figure 1 which states 7-9 hours. Please rectify as appropriate.

Line 364: "The reliability of such an approach..."

Supplemental data 1 table: Please change stable for barn in the descriptors list

Author Response

Dear reviewer,

please, see the attached file to consult our answers.

Best regards

C. Flahaut

Reviewer 2 Report

The manuscript by Daher et al. describes an interesting proteomics study to corelate sensory characterization of casein protein hydrolysates to peptide abundance feature of these hydrolysates.  In this article, sixteen enzymatic hydrolysates were first subjected to a sensory analysis to generate qualitative characteristics. The same sample sets were then analyzed by LC-MS analysis to produce peptide abundance data. Following PCA analysis, the author found the principle component in PCA analysis of LC-MS data could be used to predict the sensory characteristics of the hydrolysates. 

The major strength for this manuscript is the subject of this study.  The topic and the methodology used in this manuscript are very interesting and intriguing. The method could be very useful if it can be validated by large number of tests. The major weakness is authors don't go far enough in the validation of their findings.  The results presented are a little bit too preliminary and more validation data is clearly needed to strengthen this paper. Overall, this is a very interesting study and the manuscript is well organized. However, clarification of the questions below would help strengthen the paper.

(1) Table 1 need to be better presented. The superscript letters (a,b c,…) need to be specifically explained  

(2) The profile of detected 208 peptides need to described in more details: for example, the mass range distribution (Da) of the distribution of these peptides, the length (number of amino acid) in each peptides The major concern here is the detected peptides may be biased due to the LC-MS method and may not represent the major components in hydrolysates. It is commonly known LC-MS has difficulties to detect small (<4 amino acid) and large (>20 amino acids) peptides, which could have more impact on the sensory characteristics of the hydrolyze.

(3) In addition, the authors only considered the peptide abundance for PCA analysis. Have the authors considered adding peptide features (such as length, hydrophobicity score, retention time) into the analysis?

Author Response

(The authors gave the same response as above.)

Reviewer 3 Report

I really enjoyed this interesting manuscrupt. As new instrumentation and analytical techniques have invaded nearly every field and industry, they somehow seem to miss food science. The design of this study with LC-MS/MS and sensory evalution is creative and interesting.

However, I do think that some work is required to make this manuscript ready for publication. There are two big things missing here. 

1) Results. 

A lot of work went into the analysis of these results, but they do not seem to be in the manuscript or the supplemental information. Please provide summaries of the results from Progenesis QI and Peaks as supplemental information. 

2) Data accessibility.

Please upload the instrument files in their vendor proprietary format or as a universal format (mzmL/mzXML) and processed data to a public repository via a ProteomeXchange (http://www.proteomexchange.org/) partner such as PRIDE or MASSIVE. If this has been done, please clearly note the data availability in the text. 

This really is an interesting study and will be of much higher impact when the results are made available to other researchers. 

Minor notes: 

Lines:

51, language is a bit unclear. 

54, please provide a reference for allergenicity statement

364-372, This may fit better in the discussion section than the results section

Author Response

Dear reviewer,

please, see the attached file to consult our answers.

Best regards

C. Flahaut

This manuscript is a resubmission of an earlier submission. The following is a list of the peer review reports and author responses from that submission.

Round 1

Reviewer 1 Report

This manuscript relates to a well constructed study on the ability of high res. HPLC- MS/MS to identify the bitterness of milk protein hydrolysates. Despite a promising objectives, the wish by the authors to disclose only very loosely their hydrolysis protocols and the nature of the identified peptides, coupled with some looseness at times in the English language. Also, maybe most importantly, the authors could have confirmed the usefulness of their new found results by predicting the taste profile (as well as bitterness) of a new unknown hydrolysate, thereby validating some of their claims.

I detail below some elements that would also need addressing in the text.

Introduction. Some of the literature cited is quite old and possibly obsolete. While it could be interesting to develop an introduction using historical documents in the context of a systematic review on the subject, this is not the purpose of an introduction of a research paper. The absence of some recent articles in the field of milk peptide mapping by Mass Spectrometry for example should be rectified

Figure 1 While this figure is very nice and explain very well the purpose of the manuscript, its caption is very detailed and spans 23 lines of text. This maybe an editorial decision in the end, but these details should maybe be more appropriately placed in the body of the text.

Line 159-160 10% Nitrogen or 10% Protein? please clarify

Line 163 How is hydrolysis monitored by the degree of hydrolysis? Please explain further.

Line 165 What are the remaining major number of hydrolysates?

Line 176 and following. Were the 21 taste/flavour descriptors assigned to the panel or decided by the panel during testing. Would the authors please explain the descriptors and their choice a bit more

Line 181 “stable” Please clarify the meaning of this attribute. In English, this could mean “constant” or (more likely in your case) “horse stall, barn”

Line 189 What was the concentration of hydrolysates in the 20ml samples served?

Line 191 and others Data was analysed

Line 200 Would the authors please explain the meaning of “ peptides were concentrated (x25)”. Is this prior to SPE or a consequence of SPE, i.e. 5ml of initial hydrolysates were collected in 200ul water after SPE?

Line 215 “in two times”meaning “in duplicate”?

Line 255:Citing two 30 years old reviews as the sole source of information for the hydrolysis protocols is quite weak. While the reader can understand that the authors want to preserve some Intellectual Property on the design of their study, some more detailed information should be given, all the more so considering that the authors state using commercial enzymes to perform their hydrolysis. If not, the unlikelihood for other researchers to build from this study and develop further this field of science would render this article meaningless to the scientific community

Figures 1 a and b seem to be inverted in the caption + 1 Hydrolysate is not labelled on PCA lower right corner

Table 1 is too big for paper size and cannot be read entirely. The degree of precision on the data presented could be cut down from 3 digits after decimal point to one or two maybe? What is the variation on these data points?

Line 294 Is this the first mention of quality Controls (QCs)  in this manuscript? What are they? Please explain in the Material and Methods section

Lines 300-305 Does this mean that only 208 peptide signals were identified over the 16 hydrolysates, or just 13 per hydrolysate on average? While I can understand the authors do not want to divulge the exact sequence of these 208 peptides, more clarity could be made of the data treatment of such a small pool of peptides, for example by explaining better what region on each casein protein the peptides come from etc. Also, if only 206 peptides are identified overall, would the authors please give some detail on the robustness of the statistical treatment that gave them 5 PCAs. It is not clear in particular if the PCA analysis is done on the variance of peptide sequences or variance in peak intensity or both.

Lines 317-320 I do not understand the statistical treatment done on “the most abundant identified peptides displaying the highest mean in the hydrolysate H3”. How many peptides were taken into account here and How come is H3 on figure 3b? What is the need for this additional statistical treatment as it does not seem to lead to a different outcome. Last, why take H3 as “model hydrolysate when “H3 has a milk/fermented milk odor” while “H1 is bitter with a persistence of the bitterness” as stated line 265

Lines 347-391 These paragraph read more like an introduction than a discussion as no mention is made of the results presented in this work. The rest of the discussion reads like a repeat of the same points discussed in the results section and do not appear to bring any new angle to the discussion.

Reviewer 2 Report

The topic covered by the paper is interesting together to the presented results. The paper is well organized and easy to read, however I think come major revisions are necessary, mainly concerning the statistical analysis.

More details are reported hereafter:

  • If I well understood, the samples were tested by the assessors in duplicate. Did the order of presentation of the samples to each assessor remain the same during the two tests? For avoiding memory effects the 16 samples should have been presented to the assessors in a different order for the two tests.
  • Results bout the repeatability of the assessors should be included, at least as supplementary information.
  • It is not clear how the authors treated the data for PCA in paragraph 3.1.1. Since evaluations of each sample was available in duplicate for each sample, did the authors used average values calculated on all the assessors to calculate PCA? In this case, however, a lot of information is lost: usually in sensory evaluations, data are treated by three-way PCA (the three modes being, assessors, samples and characteristics) or by making an unfolding of one of the three dimensions: e.g. samples x assessors on the rows and characteristics on the columns. In this way, considerations on the assessors can also be carried out (repeatability, groups of similar assessors, possible presence of outliers etc.). At least a PCA by unfolding of one of the three dimensions should be provided.
  • More details about the treatment of sensory data should be provided: it is not clear which range scale was available for each characteristic to the assessors (1-5? 1-10? Other?). Moreover, it is not clear if assessors were checked for reliability (e.g. by providing them samples with extreme values of the range scale for each parameter to be evaluated etc.). This evaluation is deeply connected to how data can be scaled before PCA. In fact, it is not clear if data were centered, autoscaled or other. Autoscaling could cause an alteration to the range used by each assessor, particularly if no information is available about the assessors’ reliability.
  • Table 1: it is not clear what the letters in the table refer to, please add a description in the text and/or in the legend. Which range the intensities indicated in the table refer to? Are three decimals significant? If these values indicate the average values given by the 16 assessors, then a standard deviation should be reported together to the proper number of significant digits. Results given by the 16 assessors for all the samples and for both replicates should be reported as supplementary information.
  • Figure 3 a and b: if I well understood, the PCA in figure 3a is not based on all the identified peptides but just on the most abundant: how were they selected? Were the data normalized, centered or autoscaled? It is not clear even how the peptides were selected to be included in the PCA reported in figure 3b: did the authors use a threshold? How was it selected?
  • Table 2: please report the coefficients found for each relationship tested (PC1 vs milk, PC1 vs mild etc). Did the authors tried other tools to verify the relationship between sensory evaluations and analytical characterization based on multivariate statistics? Multifactor analysis or multiway procedures could be applied to this purpose, or also simply a PCA reporting all the data contemporarily.

 Minor comments regard some typos errors:

Line 48: change “minimize” to “minimizes”

Line 51 change “off-flavors” to “off-flavors.”

Figure 1, sensory analysis, last rectangle, please change ACP to PCA

Line 220: change “hydrolysates3.2.1. ANOVA analyses.” to

“hydrolysates.

3.2.1. ANOVA analyses.”